# Breaking the Feedback Loop of β-Cell Failure: Insight into the Pancreatic β-Cell’s ER-Mitochondria Redox Balance

**DOI:** 10.3390/cells14060399

**Published:** 2025-03-08

**Authors:** Amira Zaher, Samuel B. Stephens

**Affiliations:** 1Fraternal Order of Eagles Diabetes Research Center, University of Iowa, Iowa City, IA 52246, USA; amira-zaher@uiowa.edu; 2Department of Internal Medicine, Division of Endocrinology and Metabolism, University of Iowa, Iowa City, IA 52246, USA; 3Department of Anatomy and Cell Biology, University of Iowa, Iowa City, IA 52246, USA

**Keywords:** beta-cell function, insulin, proinsulin, ER redox, mitochondria, NAPDH, thioredoxin, protein folding

## Abstract

Pancreatic β-cells rely on a delicate balance between the endoplasmic reticulum (ER) and mitochondria to maintain sufficient insulin stores for the regulation of whole animal glucose homeostasis. The ER supports proinsulin maturation through oxidative protein folding, while mitochondria supply the energy and redox buffering that maintain ER proteostasis. In the development of Type 2 diabetes (T2D), the progressive decline of β-cell function is closely linked to disruptions in ER-mitochondrial communication. Mitochondrial dysfunction is a well-established driver of β-cell failure, whereas the downstream consequences for ER redox homeostasis have only recently emerged. This interdependence of ER-mitochondrial functions suggests that an imbalance is both a cause and consequence of metabolic dysfunction. In this review, we discuss the regulatory mechanisms of ER redox control and requirements for mitochondrial function. In addition, we describe how ER redox imbalances may trigger mitochondrial dysfunction in a vicious feed forward cycle that accelerates β-cell dysfunction and T2D onset.

## 1. Introduction

Pancreatic β-cells serve an essential physiological role in the regulation of whole animal nutrient metabolism via coordinating release of insulin with changes in blood glucose. Insulin secretion from β-cells is triggered by intracellular signals derived from glucose metabolism that promote Ca^2+^-dependent exocytosis of insulin-containing secretory granules [1,2]. Insulin release into circulation promotes nutrient uptake and storage in peripheral tissues, including liver, adipose, and skeletal muscle. Due to insulin’s singular role as the only hormone capable of lowering blood glucose, defects in β-cell function directly contribute to the onset of major forms of diabetes [3,4].

Insulin is generated via endoproteolytic cleavage of the prohormone precursor, proinsulin, which makes up 50% of all proteins synthesized in the β-cell [5,6,7]. Proinsulin is produced as a single polypeptide that contains three distinct regions, termed the B-chain, A-chain, and C-peptide (Figure 1) [8]. Following synthesis and translocation into the endoplasmic reticulum (ER), successful folding of proinsulin requires the formation of three disulfide bonds. These occur between the A and B chains and are derived from six non-sequential cysteine residues: CysA6-CysA11, CysA7-CysB7, and CysA20-CysB19 [8]. To correctly form proinsulin’s three disulfide bonds, ER protein disulfide isomerases (PDIs) and ER oxidoreductases catalyze iterative cycles of oxidation, reduction, and re-oxidation (i.e., isomerization) [8,9,10]. These disulfide bonds are critical for stabilizing proinsulin’s tertiary structure and essential for efficient ER export of proinsulin [11]. Consequently, the oxidative and reductive capacities of the ER must be carefully balanced to assist PDIs and ER oxidoreductases in both forming native disulfide bonds and eliminating non-native bonds [12]. Of note, approximately 20% of proinsulin is misfolded in the β-cell ER [13]. Thus, even in a non-pathological state, high levels of proinsulin synthesis can exceed the ER oxidative folding capacity. To combat this, homeostatic mechanisms have evolved to eliminate terminally misfolded proinsulin (and other proteins). The ER transmembrane cofactor protein, SEL1L, and the ER resident E3 ubiquitin ligase, HRD1, facilitate ER export and targeted degradation of misfolded proteins via the cytosolic proteasome in a process termed ER-associated degradation (ERAD) [14]. Importantly, complete disulfide bond reduction is required for retrotranslocation of proteins targeted for ERAD, which is mediated by the ER oxidoreductase, ERdj5 [15]. Thus, adequate reductive capacity within the ER is essential for not only correcting non-native disulfide bonds, but also clearance of misfolded proteins to preserve ER health. Collectively, these observations highlight the importance of regulating ER redox homeostasis for optimal insulin production and β-cell function.

Oxidative stress is a universal challenge to a wide array of cell populations including cardiomyocytes, lung fibroblasts, hepatocytes, and even cancer cells [18,19]. Importantly, β-cells are particularly vulnerable to oxidative stress due to their limited antioxidant capacity, as demonstrated by the low expression of antioxidant genes such as catalase (*CAT*), glutathione peroxidases (*GPX1-8*), and superoxide dismutases (*SOD1, SOD2*) [20]. Instead, β-cells heavily rely on the peroxiredoxin/thioredoxin system as their primary antioxidant defense mechanism [21]. The attenuated antioxidant defense of the β-cell may be critical for the utilization of metabolism-derived oxidants, such as hydrogen peroxide and nitric oxide, and reductants, nicotinamide adenine dinucleotide phosphate (NAPDH) and glutathione (GSH), in promoting insulin release [21,22]. Recently, metabolic redox cycles involving NAPDH, glutathione, and thioredoxin were also linked to β-cell ER function and insulin production [23]. Hyperoxidation of the ER lumen was identified as a characteristic feature of β-cell failure that, when reversed, restored proinsulin trafficking and insulin secretion [23]. Moreover, mitochondrial dysfunction directly promoted ER hyperoxidation whereas alleviating mitochondrial stress restored ER redox homeostasis [23]. Taken together, these observations point towards crosstalk between the ER and mitochondria in the context of β-cell ER redox regulation relevant to insulin production. Whether this relationship between ER redox and mitochondrial function is bidirectional remains unknown. In this article, we will discuss aspects of ER redox regulation, the role of mitochondrial dysfunction in ER redox alterations, and present a framework for ER redox acting as a regulator of mitochondrial function.

## 2. β-Cell ER Redox Control

The oxidizing environment of the ER lumen facilitates protein folding through a complex interplay of chaperones, PDIs, and ER oxidoreductases. This folding process is assisted by the formation of disulfide bonds, which help to constrain protein tertiary structure into native, low-energy conformations. In the proinsulin primary sequence, 6 cysteine residues can give rise to 15 possible disulfide bond configurations. Given the proclivity of proinsulin to misfold [13], the formation of non-native disulfide bonds seems not only probable, but likely a frequent step in proinsulin maturation. To overcome cysteine mispairings, the oxidative capacity of the ER must be carefully balanced with reductive mechanisms to ensure either resolution of non-native disulfide bonds, or, in the event of terminal misfolding, to target proteins for ERAD. Under chronic metabolic stress, inflammation, or redox insults, perturbations to ER redox balance can occur, leading to excess generation of reactive oxygen species (ROS), irreversible protein oxidation, and accumulation of misfolded proteins. In β-cells, recent studies demonstrate that ER hyperoxidation leads to the accumulation of non-native proinsulin aggregates, which limit the proinsulin supply available for insulin granule formation [23,24,25]. In the following section, we discuss the key mediators of β-cell ER redox regulation and possible scenarios for how redox imbalances develop.

### 2.1. ERO1

ER oxidoreductin-1 (ERO1) is a highly conserved enzyme that plays a key role in oxidative protein folding, where it regenerates oxidized PDIs to ensure continuous disulfide bond formation in nascent proteins [26,27,28]. ERO1 acts as an electron acceptor that transfers electrons from reduced PDIs to its active site containing flavin adenine dinucleotide (FAD), which subsequently reduces molecular oxygen to form hydrogen peroxide [26,28,29]. Consequently, ROS, particularly hydrogen peroxide (H_2_O_2_), is a byproduct of ER oxidative protein folding and contributes to the ER oxidative potential (Figure 2). The yeast homolog Ero1p is tightly self-regulated through oxidation of two non-catalytic cysteines that modulate Pdi1p access to the active site loop containing four catalytic cysteines [30]. This critical feedback mechanism adjusts Ero1p activity to the demands of oxidative protein folding. The predominant mammalian isoform, ERO1α, employs a similar self-regulatory mechanism, but through disulfide bond bridges between active site cysteines (Cys94 and Cys99) and non-active site cysteines (Cys131 and Cys104). Formation of the regulatory disulfide bonds in both Ero1p and ERO1α are essential for preventing overproduction of hydrogen peroxide and limiting ER hyperoxidation.

The β-cell-specific ERO1 isoform, ERO1β, plays a critical role in maintaining insulin secretion and glucose homeostasis by regulating ER redox balance. Loss of ERO1β resulted in the accumulation of misfolded proinsulin, reduced islet insulin content, and hyperglycemia in mice [31]. Similarly, transcriptional suppression of ERO1β in insulin-producing MIN6 cells significantly decreased insulin content and secretion [32]. Despite the conservation of shared regulatory mechanisms with ERO1α, ERO1β exhibits a much broader range of redox activity that can elicit ER hyperoxidation in some contexts [33]. For example, ERO1β overexpression significantly increased ER hydrogen peroxide content [34] and led to impaired insulin secretion and hyperglycemia in a mouse model [35]. In addition, the cell-permeable reducing agent, dithiothreitol (DTT), was used to restore insulin secretion in ERO1β overexpressing MIN6 cells [35]. Collectively, these observations underscore the importance of the β-cell’s ER redox balance and highlight a critical role for the regulation of ERO1β activity in maintaining β-cell function.

Whether pathological activation of ERO1β contributes to β-cell dysfunction in T2D remains to be determined. Recently, hyperoxidized ER was identified in diabetes models with impaired ER export of proinsulin [23,24]. These findings correlate with the accumulation of non-native proinsulin aggregates observed in T2D human islets [36]. Potentially, persistent or excess ERO1β overactivation, such as during nutrient overload or ER stress [37,38], could overshoot the demands for oxidative protein folding and strain the ER’s reductive capacity for disulfide bond isomerization and/or clearance of misfolded proteins. As a result, ER hyperoxidation in the β-cell would impair proinsulin folding and limit the proinsulin supply available for insulin granule formation. Through this mechanism, chronic ER hyperoxidation would lead to a gradual depletion of insulin stores that, when compounded with peripheral insulin resistance, contributes to the loss of glucose tolerance and development of diabetes.

### 2.2. Peroxiredoxin 4

β-cells primarily utilize the peroxiredoxins (PRDX) for antioxidant scavenging rather than GPXs, SODs, or CAT, which are highly expressed in other tissues [20,21]. Here, we will focus on the ER resident PRDX4, but note, GPX7 and GPX8 are also ER-localized oxidant scavengers yet these enzymes are not expressed in β-cells [39]. PRDX4 is a classic 2-Cys peroxiredoxin that uses cysteine dithiols to neutralize hydrogen peroxide via formation of sulfinic acid [40,41,42] (Figure 2). Resolution occurs by reacting with an adjacent PRDX4 protein or sulforedoxin activity. In addition, PRDX4 can functionally replace ERO1 in the generation of hydrogen peroxide and promote PDI oxidation for client protein disulfide bond formation [41,43]. These observations highlight the bidirectional ability of PRDX4 to both scavenge and produce oxidants; however, PRDX4’s oxidative capacity is likely to be less efficient given the β-cell’s dependence on ERO1β for normal ER redox homeostasis [31,38].

In β-cells, PRDX4 overexpression was shown to increase proinsulin biosynthesis and insulin secretion and protect mice from STZ-induced diabetes [44,45]. Furthermore, silencing PRDX4 results in the accumulation of misfolded proinsulin, whereas overexpressing PRDX4 reverses this phenotype [46]. Thus, unlike ERO1β, PRDX4 largely correlates with improved β-cell function. Despite this benefit, PRDX4 is not immune to failure. Persistent exposure to hydrogen peroxide leads to PRDX4 hyperoxidation by converting the cysteine thiols from sulfinic acid to sulfonic acid. This oxidation state cannot be reduced by the resolving cysteine and leads to the irreversible inactivation of PRDX4 [47,48]. In models of β-cell dysfunction, accumulation of the inactive hyperoxidized PRDX4 has been observed [46], which correlates with hyperoxidation of the ER and proinsulin misfolding [23,24]. Whether PRDX4 hyperoxidation in diabetes models is due to excess ERO1β activity is not known, but as discussed earlier, ERO1β overexpression promotes ER hyperoxidation, which consequently leads to inactivation of PRDX4 [40]. Thus, loss of the protective mechanisms afforded by PRDX4 to maintain ER redox balance may be a critical determinant in the development of β-cell dysfunction in diabetes.

### 2.3. Thioredoxin

Oxidation of PDIs to facilitate disulfide bond formation is regulated by ERO1 activity [26,27,28]. Of equal importance, PDIs can also function as reductases to eliminate non-native disulfide bonds [49], which readily form in proinsulin [25,36], yet the mechanism and source of reductants for PDIs is not clear [50]. One potential source is provided by the influx of nascent proteins translocating into the ER that contain reduced cysteine thiols. In the case of the β-cell, there are six cysteines per proinsulin molecule synthesized that could substantially contribute to PDI reduction. In addition, prominent cellular redox carriers, glutathione and thioredoxin, which are regulated metabolically, have also been identified as important ER redox buffers [23,50,51,52]. While glutathione was identified as the primary ER reductant in hepatocytes [51], the unique demands of proinsulin folding in the β-cell may favor the reductive potential of thioredoxin [23]. Based on these recent studies [23], we will focus on the regulation of thioredoxin as a β-cell ER reductant.

Thioredoxin is a small (12–14 kDa), ubiquitously expressed cytosolic protein that plays a central role in cellular antioxidant defense [53,54] (Figure 2). Thioredoxin contains a dithiol/disulfide active site that is used as a redox carrier by peroxiredoxins and other antioxidant proteins to scavenge peroxides and methionine sulfoxide and also reduce protein disulfide bonds [54]. Following electron transfer, oxidized thioredoxin is recycled by the cytosolic NADPH dependent enzyme, thioredoxin reductase-1 (TXNRD1). In β-cells, overexpression of thioredoxin was shown to protect against STZ-induced diabetes in mice, attenuate the development of glucose intolerance from high fat diet feeding, and enhance islet engraftment with improved glycemic control in diabetic NOD mice [21]. More directly, overexpression of thioredoxin protected against hydrogen peroxide induced toxicity [55] and treatment with thioredoxin mimetics restored insulin secretion that was impaired by the TXNRD1 inhibitor, auranofin [56]. Under diabetogenic β-cell stresses, such as hyperglycemia, the endogenous thioredoxin inhibitor, thioredoxin interacting protein (TXNIP), is upregulated [57,58]. TXNIP binds to and sequesters reduced thioredoxin and thereby limits the β-cell’s antioxidant defense. Because TXNIP is strongly associated with impaired insulin secretion and β-cell apoptosis in diabetes models [59,60,61], inactivation of TXNIP is being pursued clinically for both T1D and T2D treatments [62].

Thioredoxin has emerged as a critical regulator of β-cell ER redox homeostasis. A recent study revealed that genetic knockout of *Txnrd1* in β-cells strongly impaired insulin secretion and led to upregulation of compensatory antioxidant pathways [63]. More directly, pharmacological inhibition or genetic suppression of TXNRD1 resulted in ER hyperoxidation and impaired proinsulin exit from the ER [23]. In contrast, suppression of TXNIP restored ER redox homeostasis and partially recovered insulin secretion in diabetic β-cell models [23]. Collectively, these data highlight the importance of the thioredoxin pathway to β-cell ER redox control. Tentatively, non-native disulfide bonds that readily form in proinsulin may be sufficiently stable that reduction and isomerization requires specific ER reductases, such as ERdj5, which operate independently of the glutathione redox system [50]. In addition, ERdj5 has a critical role in protein disulfide bond reduction for retrotranslocation of misfolded proteins during ERAD [15]. This requirement for resolving non-native disulfide bonds in proinsulin would explain the β-cell’s preference for thioredoxin as the preferred ER reductant. A remaining question surrounds the observation that unlike glutathione, thioredoxin does not directly enter the ER (Figure 2). Thus, how cytosolic thioredoxin contributes to the resolution of non-native disulfide bonds in the ER lumen is not clear. In prokaryotes, thioredoxin is restricted to the cytoplasm yet is critical for protein disulfide bond reduction within the periplasmic space. Electron transfer occurs through reduction of the membrane protein DsbD, which shuttles electrons to the periplasmic protein disulfide isomerase, DsbC, for disulfide bond exchange [64]. In mammalian cells, recent studies demonstrate that a similar electron shuttle occurs between cytosolic thioredoxin and an ER membrane protein to support disulfide bond reduction in the ER lumen [65]; however, the identity of the membrane protein and distal steps for transfer to PDIs, such as ERdj5, have yet to be defined.

## 3. Mitochondrial Regulation of ER Redox Homeostasis

Recent studies demonstrate that oxidative protein folding in the ER is regulated via a metabolic redox relay through the supply of NAPDH, glutathione, and thioredoxin [23,50,51,52]. As previously discussed, thioredoxin and glutathione can assist ER resident PDIs in forming native disulfide bonds and reducing non-native bonds, which has been referred to as redox buffering [23]. Importantly, recycling oxidized glutathione and thioredoxin to their reduced states is catalyzed by cytosolic NADPH-dependent reductases, glutathione reductase (GSR1) and TXNRD1, respectively. In turn, the terminal reductant, NADPH, has been shown by multiple studies to precisely fluctuate with changes in β-cell glucose metabolism [66,67,68,69]. These observations directly link metabolic activity in the β-cell with available redox donors to support ER oxidative protein folding.

Several enzymes contribute to NADPH reduction in β-cells, including cytosolic isocitrate dehydrogenase-1 (IDH1), mitochondrial IDH2, and glucose-6-phosphate dehydrogenase (G6PDH). These enzymes are required for insulin secretion, but not mitochondrial function [69,70,71,72,73]. Additional NADPH-generating enzymes involved in one-carbon folate metabolism can also influence β-cell function, including methylenetetrahydrofolate dehydrogenase 2 (MTHFD2) and aldehyde dehydrogenase 1 family member L2 (ALDH1L2), yet their contribution to the NADPH pool is less clear [74]. While malic enzymes (four isoforms) can also generate NADPH, their loss does not affect glucose homeostasis or insulin secretion [75], suggesting only a minor role in β-cell function.

In support of the metabolic redox relay, suppressing NADPH cycling in mouse islets via IDH1 knockdown disrupted ER redox and impaired proinsulin trafficking [23]. This impairment was reversible using the reducing agent, DTT, highlighting the importance of NADPH-derived reductants in ER redox homeostasis [23]. Furthermore, the mitochondrial-directed antioxidant, MitoQ, restored NADPH cycling and ER redox balance in diabetic β-cell models [23]. While the role of other NADPH-producing enzymes discussed above in β-cell ER redox control remains uncertain, IDH2 suppression in hepatocytes has also been linked to ER redox defects [51].

The pathways linking metabolic and ER functions in β-cells are important for three reasons. First, glucose metabolism is primarily directed through mitochondrial oxidative phosphorylation to generate adenosine trisphosphate (ATP) along with signaling intermediates known to regulate insulin exocytosis [2]. Absence of lactate dehydrogenases (LDH A, B, C) and the monocarboxylate carrier (SLC16A1) to export lactate prevents the β-cell from utilizing anaerobic glycolysis as an alternate mechanism for energy and substrate production [76]. As a result, β-cells depend on mitochondria for both ATP production and maintaining cellular redox balance. Second, the β-cell is uniquely adapted to sense physiological fluctuations in glucose because of the K_m_ of the glucose transporters, GLUT1 and GLUT2, and glucokinase, which operate within the physiological range of blood glucose, 4.4–6.1 mM [2,77,78]. Consequently, β-cell metabolism, including redox cycles, oscillates with increased activity during the fed state and decreased activity in the fasted state [66,67,68,69]. Moreover, glucose metabolism not only triggers insulin secretion, but also stimulates *INS* mRNA translation [79,80,81]. Thus, the regulation of multiple tiers of the insulin biosynthetic and exocytic pathways are efficiently coordinated by metabolic cycles [1]. Finally, mitochondrial dysfunction and declined secretory capacity are hallmarks of β-cell dysfunction in T2D; however, their mechanistic connection has only recently been established [23]. Depletion of cellular ATP stores can certainly strain ER functions (e.g., protein folding, Ca^2+^ uptake, ERAD), but in addition, redox imbalances also adversely affect oxidative protein folding (e.g., disulfide bond formation). Indeed, restoring ER redox balance with exogenous reductants or hydrogen peroxide scavengers can promote proinsulin trafficking even when mitochondrial function is compromised [23]. Similarly, glucokinase inhibition rapidly led to ER hyperoxidation, which can be reversed by supplementation with cellular reductant [23]. Thus, loss of mitochondrial function and the concomitant depletion of the redox relay can negatively impact proinsulin folding in the ER and thereby limit insulin production.

## 4. Trading Places: Can ER Redox Alter Mitochondrial Function in β-Cells?

The ER and mitochondria are highly integrated organelles that orchestrate multiple cellular functions and metabolic processes, including lipid metabolism, calcium signaling, energy metabolism, and redox homeostasis. The regulatory relationship between these two organelles is established, in part, through mitochondria-ER contact sites (MERCs) [82]. MERCs typically span 10–30 nm between the ER and outer mitochondrial membrane and connect via tethering proteins, Inositol 1,4,5-trisphophate receptors (IP_3_R), voltage-dependent anion channels (VDAC), glucose-regulated protein-75 (GRP75), Parkinson disease protein (DJ1), and Mitofusin 2 (MFN2) [83]. These contact sites enable bidirectional communication that enhance both mitochondrial ATP production and ER oxidative protein folding. In β-cells, disruptions to mitochondrial function lead to ER redox imbalances that strain the proinsulin folding machinery [23]. Whether ER redox imbalances also feedback to disrupt mitochondrial functions is not known but could represent an unexplored yet potentially critical area of diabetes etiology. In the sections below, we propose three mechanisms through which ER-derived ROS can induce mitochondrial dysfunction: mitochondrial calcium overload, ferroptosis, and inflammasome activation. Emerging evidence suggests that all three mechanisms are diabetogenic and contribute to β-cell dysfunction.

### 4.1. Mitochondrial Calcium Overload

MERCs are subject to direct redox regulation by ROS through the presence of redox-sensitive proteins. For example, IP_3_R can be oxidized by hydrogen peroxide, leading to enhanced Ca^2+^ transport from the ER to the mitochondria through VDAC [84,85]. We propose that the accumulation of ER hydrogen peroxide from excess ERO1β activity leads to IP_3_R oxidation (Figure 3). In addition, Protein Kinase R-like Endoplasmic Reticulum Kinase (PERK) is also subject to hydrogen peroxide dependent oxidation, which promotes PERK oligomerization as an additional MERC tethering complex [86,87]. Consequently, PERK oxidation exacerbates the effects of IP_3_R oxidation by amplifying ER-mitochondrial Ca^2+^ transport. While Ca^2+^ influx into the mitochondria can promote ATP production [88], excess mitochondrial Ca^2+^ disrupts oxidative phosphorylation, leading to ATP depletion and ROS overproduction [89]. Through this mechanism, a feedforward cycle of oxidative stress and mitochondrial dysfunction is triggered.

In the development of T2D, the extent of MERC tethering and/or the number of MERCs varies between stages of ER stress and β-cell dysfunction [90]. For example, acute glucose stimulation of human islets increased ER-mitochondrial tethering and Ca^2+^ exchange [91], similar to acute activation of ER stress with tunicamycin [86]. Tentatively, this protective mechanism operates to improve organelle bioenergetics and functions. In contrast, islet studies using chronic culture with elevated glucose demonstrated depletion of ER Ca^2+^ stores and increased mitochondrial Ca^2+^ that coincided with defects in mitochondrial function, ATP generation, and overall diminished glucose-stimulated insulin secretion (GSIS) [91]. Thus, excess ER-mitochondria Ca^2+^ exchange can contribute to organelle dyshomeostasis and damage. In support of this notion, mitochondrial calcium overload through IP_3_R was reported in human T2D patients, mouse models of diabetes, and cultured β-cells [92]. In these advanced models of β-cell dysfunction, MERC numbers are diminished, which further highlights the complex and dynamic regulation of MERC assembly and function in the β-cell response to metabolic perturbations.

### 4.2. Ferroptosis Induction

Beyond regulating Ca^2+^ mobilization through MERCs, the effects of ER-generated ROS also extend to lipid metabolism. Through Fenton chemistry, iron and hydrogen peroxide generate hydroxyl radicals that can react with and damage membrane lipids, nucleic acids, and proteins [19]. Lipid peroxidation, a key process in ferroptosis, occurs as polyunsaturated fatty acids (PUFAs) are oxidized in the presence of hydrogen peroxide and redox-active iron [93]. Lipid radicals are initially produced that go on to react with molecular oxygen, which form lipid peroxyl radicals. These lipid peroxides can also generate reactive aldehyde derivatives, such as 4-hydroxynonenal (4-HNE) and malondialdehyde (MDA) [94,95]. Collectively, these lipid-derived radicals significantly alter mitochondrial morphology, evidenced by altered membrane fluidity, shrinking or enlarged cristae, mitochondrial fragmentation, and mitochondrial DNA release [93,96,97,98,99].

Strong evidence links ferroptosis to β-cell failure and death following a variety of diabetogenic stressors, including high glucose culture, STZ, hydrogen peroxide, and proinflammatory cytokines, which all accumulate ROS, lipid peroxides, and redox-active iron [100,101]. Importantly, β-cell function and viability can be restored in these models using a ferroptosis inhibitor, ferrostatin, whereas ferroptosis inducers, such as erastin and RSL3, are sufficient to compromise viability and GSIS in human islets [102]. Based on these data, we propose that hydrogen peroxide generated by overactive ERO1β diffuses from the ER into the mitochondria, where it promotes lipid peroxidation and ferroptosis (Figure 3). Mitochondrial membranes undergo fragmentation and shrinkage, decreasing ATP generation and increasing ROS production. Thus, excess ROS produced from ER oxidative folding can potentially damage mitochondrial membranes and cripple mitochondrial function through ferroptosis. As ferroptosis progresses, β-cell viability declines, compounding insulin secretion deficits with peripheral insulin resistance and hyperglycemia.

### 4.3. NLRP3 Inflammasome Activation

The NOD-, LRR-, and pyrin domain-containing protein-3 (NLRP3) inflammasome has been implicated in β-cell failure in both T1D and T2D. This multiprotein complex responds to microbial infection and cellular damage by initiating inflammatory responses through the activation of caspase-1, which cleaves proinflammatory cytokines, such as interleukin-1β (IL-1β) and interleukin-18 (IL-18), into their active forms [103]. In T2D, chronic low-grade inflammation sustained by NLRP3 activation exacerbates insulin resistance in peripheral tissues and decreases β-cell mass via IL-1β-mediated apoptosis and pyroptosis [104]. A key mechanism underlying NLRP3 activation is oxidative stress, where excessive ROS serves as a molecular trigger. While mitochondrial ROS is thought to be the primary activator of NLRP3 [105,106], ER-derived ROS may also directly activate the NLRP3 inflammasome (Figure 3). The localization of NLRP3 to MERCs under oxidative stress highlights this functional link between the ER and mitochondria in inflammasome regulation. Further supporting this pathway, the ER-resident NADPH oxidase-4 (NOX4), which generates hydrogen peroxide, can enhance NLRP3 activation, whereas suppression of NOX4 attenuates inflammasome activity [107]. We propose that an imbalance in ER redox homeostasis through excess ERO1β activity and concomitant inactivation of PRDX4 also lead to hydrogen peroxide activation of the NLRP3 inflammasome. In addition, mitochondrial calcium overload via IP_3_R oxidation, described above, can activate NLRP3 [108]. Collectively, these findings suggest that multiple ER-derived oxidative signals, in conjunction with mitochondrial dysfunction, act as key drivers of NLRP3 inflammasome activation. This sustained inflammasome activity can contribute to β-cell damage, impairing glucose sensing and insulin secretion, and thereby accelerating diabetes pathogenesis.

### 4.4. A Model for ER-Directed Mitochondrial Dysfunction

In addition to proinsulin folding defects, we propose that ER redox imbalance plays a critical role in β-cell dysfunction by disrupting mitochondrial homeostasis through the three major pathways described above: calcium overload, ferroptosis, and NLRP3 inflammasome activation (Figure 3). Excessive ROS from ER oxidative protein folding leads to the oxidation of IP_3_R and PERK and thereby increases Ca^2+^ transport to the mitochondria. The calcium overload disrupts oxidative phosphorylation, depletes ATP, and enhances ROS production. Simultaneously, hydrogen peroxide from the ER interacts with iron to drive lipid peroxidation through ferroptosis, which severely damages mitochondrial membranes and compromises mitochondrial functional integrity. In parallel, ER-derived ROS also activate the NLRP3 inflammasome, which promotes chronic inflammation. Collectively, these interconnected mechanisms conspire to create a feedback loop of oxidative stress, mitochondrial dysfunction, and inflammation that ultimately promote β-cell failure in the development of T2D.

## 5. Concluding Remarks

During the development of T2D, insulin resistance and the resultant hyperglycemia drive an increase in proinsulin synthesis to meet rising metabolic needs. We propose that the heightened demand for insulin exceeds the ER’s redox buffering capacity, leading to hyperactivation of ERO1β, overproduction of hydrogen peroxide, and inactivation of the oxidant scavenger PRDX4. This disruption in ER redox homeostasis triggers mitochondrial calcium overload, ferroptosis, and NLRP3 inflammasome activation, which collectively impair mitochondrial function and further elevate ROS production. The resulting imbalance between ER redox homeostasis and mitochondrial function creates a vicious feed-forward cycle. Mitochondrial dysfunction deprives the ER of essential reductants, NADPH and thioredoxin, pushing oxidative protein folding machinery into a further hyperoxidized state that disrupts proinsulin maturation. As misfolded proinsulin accumulates, ER stress intensifies, while increased hydrogen peroxide production exacerbates mitochondrial dysfunction. Thus, diabetes progression is not solely driven by insulin resistance or glucose toxicity but by a fundamental failure in β-cell organelle homeostasis. Moreover, proinsulin misfolding is not merely a consequence of ER dysfunction but an active contributor to β-cell metabolic decline. A key question is whether targeted redox modulation at the ER can restore mitochondrial function, or if both compartments must be addressed independently to preserve β-cell function.

While our proposed model may help to disentangle the complex relationships and intricacies of β-cell redox homeostasis, there are notable challenges to studying redox metabolism and significant barriers that may preclude translating these findings into clinical management of T2D. For example, the large sample sizes needed for traditional biochemical and liquid chromatography-mass spectrometry based methods used in detailed redox metabolite analysis, activity of redox enzymes, etc., are often not feasible with the limited availability of cadaveric human donor islets. Furthermore, the high degree of natural biological variation and prep-to-prep variability in islet isolation can compound interpretations of analytical data. With the emergence of genetically-encoded redox sensitive probes, the limitations of human islet sample sizes can be overcome; however, the implementation of these probes requires the introduction of genetic material. More recently, the development of highly sensitive methods and instrumentation for single-cell metabolomic profiling may afford significant advancements in our understanding of β-cell redox metabolism, particularly from precious samples acquired from diabetic patients. In addition, recent strides in differentiation of human stem-cell derived β-cells and generation of islet organoids may also provide an abundant source of human tissue for the study of redox metabolism. Lastly, to bridge the gap between fundamental discoveries and clinical applications, basic findings regarding the redox homeostasis of β-cells must be expanded beyond systemic antioxidant therapies to β-cell-targeted strategies. The development of next-generation antioxidant agents with enhanced potency and specificity could provide a more effective means of mitigating oxidative stress in β-cells, potentially serving as an adjunct therapy to enhance current treatment options for T2D management. Despite these challenges, unraveling the complexities of ER-mitochondrial redox control has the potential to drive the development of precision therapies that better safeguard β-cell functions as a durable intervention in diabetes management.

## Figures and Tables

**Figure 1 cells-14-00399-f001:**
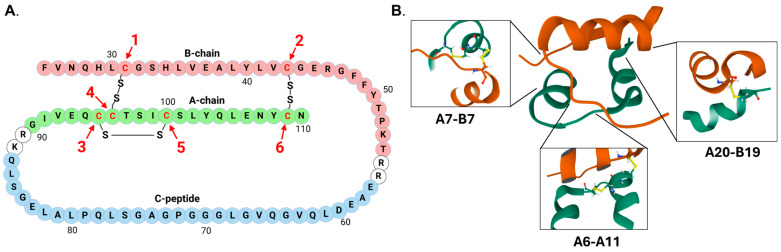
Proinsulin and insulin structure. (**A**) Linear proinsulin sequence highlighting the order of the six cysteine residues that form the three disulfide bonds. Sequence is colorized: brown is the B chain, green is the A chain, blue is C-peptide. (**B**) Structure of insulin highlighting the disulfide bonds between the A and B chains (A7-B7, A20-B19) and within the A chain (A6-A11), which are necessary for proper structure. Structure is colorized as follows: brown is the B chain; green is the A chain. Images were created using Mol* of 4EZT from the RCSB PDB [16,17].

**Figure 2 cells-14-00399-f002:**
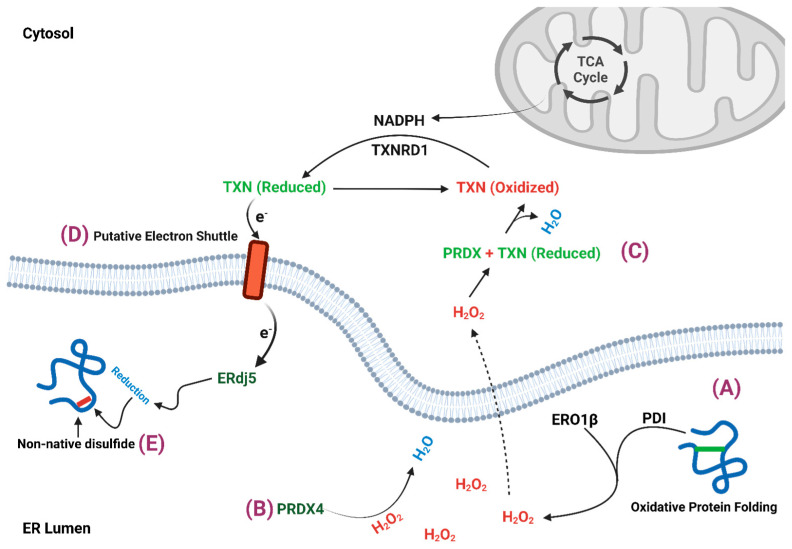
Overview of ER redox regulation. (A) Oxidative protein folding generates hydrogen peroxide (H_2_O_2_) through ERO1β. (B) Hydrogen peroxide is scavenged by PRDX4 in the ER lumen. (C) Hydrogen peroxide diffuses into the cytosol to be scavenged by cytosolic antioxidants such as peroxiredoxin with the aid of the redox carrier thioredoxin (TXN). (D) TXN is reduced by TXNRD1 and NADPH. In addition, electrons from TXN can be transferred into the ER via reduction-oxidation of an unknown ER membrane shuttle to aid ERdj5-like PDIs in the reduction of non-native disulfide bonds (E).

**Figure 3 cells-14-00399-f003:**
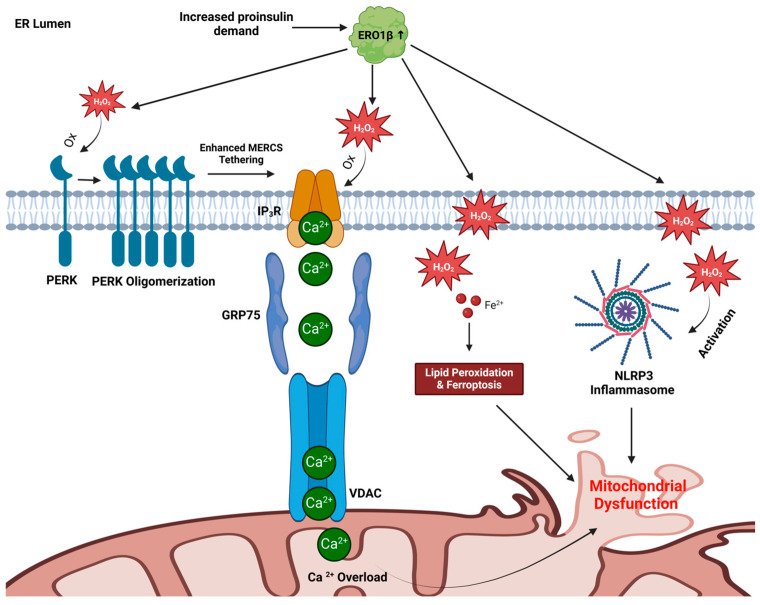
A model describing how ER hyperoxidation promotes mitochondrial dysfunction. With elevated demand for proinsulin synthesis, oxidative protein folding increases leading to excess ERO1β activity and hydrogen peroxide (H_2_O_2_) production. Hydrogen peroxide can oxidize PERK leading to its oligomerization, which enhances Mitochondrial-ER contacts (MERCs) tethering. Hydrogen peroxide can also oxidize IP_3_R and increase the transport of Ca^2+^ from the ER into the mitochondria through VDAC, leading to Ca^2+^ overload and mitochondrial dysfunction. Hydrogen peroxide can also diffuse from the ER into the mitochondria and cytosol and promote lipid peroxidation/ferroptosis and NLRP3 inflammasome activation, both of which can lead to mitochondrial dysfunction.

## Data Availability

Not applicable.

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
