# Peer review of "Breaking the Feedback Loop of β-Cell Failure: Insight into the Pancreatic β-Cell’s ER-Mitochondria Redox Balance"

_cells, 2025, doi:10.3390/cells14060399_

Round 1

Reviewer 1 Report

Comments and Suggestions for Authors

This is well-written and comprehensive review. While some aspects of the review are not particularly novel in their reporting, the review provides an important resource and commentary on the current state of ER - mitochondrial imbalance in the context of the beta-cell dysfunction that precedes type 2 diabetes development. 

I have no specific comments or suggestions for the manuscript and endorse its publication in its present form. 

Author Response

Comment 1: This is well-written and comprehensive review. While some aspects of the review are not particularly novel in their reporting, the review provides an important resource and commentary on the current state of ER - mitochondrial imbalance in the context of the beta-cell dysfunction that precedes type 2 diabetes development. 

I have no specific comments or suggestions for the manuscript and endorse its publication in its present form. 

Author response: We thank the reviewer for their support.

Reviewer 2 Report

Comments and Suggestions for Authors

This paper is a timely, well-crafted review of the role of ER and mitochondria and their bidirectional crosstalk, and what has potential to go wrong that conspires in the development of diabetes. Relatively minor issues are listed below:

Issues to address:

  1. Please make adjustments to Fig 3 so that it better represents the description of MERCs; the MERC contact sites should be depicted to support the author’s statement "MERCs typically span 10-30 nm between the ER and outer mitochondrial membrane and connect via tethering proteins, Inositol 1,4,5-trisphophate receptors (IP3R), voltage-dependent anion channels (VDAC), glucose-regulated protein-75 (GRP75), Parkinson disease protein (DJ1), and mitofusins."
  2. In the Concluding remarks section, there is a missed opportunity to reflect on limitations of the research in this area that are needed to be overcome for discriminating the molecular events, or for resolving disparate views/opinions/findings in this field. For example, are there new technologies that will be needed? If there are disparate findings, why?—are they technical in nature, or perhaps due to model systems used?
  3. The authors statement “Thus, attenuating ERO1β activity could be a useful anti-diabetogenic strategy that ameliorates β-cell exhaustion by enhancing insulin production” is a bit confusing to the reader because it follows prior text in this section describing the negative consequences of ERO1beta attenuation. If the authors could share their vision for this as a strategy in greater detail, balanced with the knowledge presented on the prior page regarding the negative consequences of ERO1b attenuation, it would provide clarity and add value to the already outstanding article.  
  4. MINOR: typographical errors on line 120 and on line 239.

Author Response

This paper is a timely, well-crafted review of the role of ER and mitochondria and their bidirectional crosstalk, and what has potential to go wrong that conspires in the development of diabetes. Relatively minor issues are listed below:

Issues to address:

Comment 1: Please make adjustments to Fig 3 so that it better represents the description of MERCs; the MERC contact sites should be depicted to support the author’s statement "MERCs typically span 10-30 nm between the ER and outer mitochondrial membrane and connect via tethering proteins, Inositol 1,4,5-trisphophate receptors (IP3R), voltage-dependent anion channels (VDAC), glucose-regulated protein-75 (GRP75), Parkinson disease protein (DJ1), and mitofusins."

Author response: This is an excellent suggestion.  We have amended Fig. 3 to better represent the significance of the ER-mitochondrial space in redox exchange, including MERCs.

Comment 2: In the Concluding remarks section, there is a missed opportunity to reflect on limitations of the research in this area that are needed to be overcome for discriminating the molecular events, or for resolving disparate views/opinions/findings in this field. For example, are there new technologies that will be needed? If there are disparate findings, why?—are they technical in nature, or perhaps due to model systems used?

Author response: This is another excellent suggestion. We have now included a section in the conclusions describing current barriers and possible means to overcome. The following text was added to Page 11, starting line 438:

“While our proposed model may help to disentangle the complex relationships and intricacies of β-cell redox homeostasis, there are notable challenges to studying redox metabolism and significant barriers that may preclude translating these findings into clinical management of T2D. For example, the large sample sizes needed for traditional biochemical and liquid chromatography-mass spectrometry based methods used in detailed redox metabolite analysis, activity of redox enzymes, etc. are often not feasible with the limited availability of cadaveric human donor islets. Furthermore, the high degree of natural biological variation and prep-to-prep variability in islet isolation can compound interpretations of analytical data. With the emergence of genetically-encoded redox sensitive probes, the limitations of human islet sample sizes can be overcome, however the implementation of these probes requires introduction of genetic material. More recently, the development of highly sensitive methods and instrumentation for single-cell metabolomic profiling may afford significant advancements in our understanding of β-cell redox metabolism, particularly from precious samples acquired from diabetic patients. In addition, recent strides in differentiation of human stem-cell derived β-cells and generation of islet organoids may also provide an abundant source of human tissue for the study of redox metabolism. Lastly, to bridge the gap between fundamental discoveries and clinical applications, basic findings regarding the redox homeostasis of β-cells must be expanded beyond systemic antioxidant therapies to β-cell-targeted strategies. The development of next-generation antioxidant agents with enhanced potency and specificity could provide a more effective means of mitigating oxidative stress in β-cells, potentially serving as an adjunct therapy to enhance current treatment options for T2D management. Despite these challenges, unraveling the complexities of ER-mitochondrial redox control has the potential to drive the development of precision therapies that better safeguard β-cell functions as a durable intervention in diabetes management.”

Comment 3: The authors statement “Thus, attenuating ERO1β activity could be a useful anti-diabetogenic strategy that ameliorates β-cell exhaustion by enhancing insulin production” is a bit confusing to the reader because it follows prior text in this section describing the negative consequences of ERO1beta attenuation. If the authors could share their vision for this as a strategy in greater detail, balanced with the knowledge presented on the prior page regarding the negative consequences of ERO1b attenuation, it would provide clarity and add value to the already outstanding article.  

Author response: We agree with this reviewer on the paradoxical nature of this statement and have removed this comment from the text. Page 7, line 150

Comment 4: MINOR: typographical errors on line 120 and on line 239.

Author response: We have fixed this error.

Reviewer 3 Report

Comments and Suggestions for Authors

In the manuscript, the authors review the mechanisms of ER redox regulation in pancreatic beta cells and report how they can be altered by mitochondrial dysfunction. Finally, based on the existing literature and their recent data, they propose that ER redox imbalances may in turn trigger mitochondrial dysfunction via calcium overload, ferroptosis and NLRP3 inflammasome activation, creating a vicious cycle that accelerates β-cell dysfunction and the onset of T2D.

The topic is relevant is relevant to the field of beta cell physiology and up to date. I have only few suggestions that can improve the text

Figure 2. Please reconsider the cartoon (not clear the Trx function in H2O2 handling) and describe panel C in the figure legend.

According to the authors’ idea and the cartoon (figure 2), trx function mainly in the cytosol and ERdj5 in the ER. Is this correct? Please better explain this point in the text and figure legend.

The authors suggest that alterations in the redox state in the ER directly affect mitochondrial function via ER-mitochondrial contact sites. However, recent data indicate that MERCs are altered (decreased) in T2D (DOI: 10.1016/bs.ircmb.2021.06.001). How can this evidence be in line with the authors' hypothesis?

Author Response

In the manuscript, the authors review the mechanisms of ER redox regulation in pancreatic beta cells and report how they can be altered by mitochondrial dysfunction. Finally, based on the existing literature and their recent data, they propose that ER redox imbalances may in turn trigger mitochondrial dysfunction via calcium overload, ferroptosis and NLRP3 inflammasome activation, creating a vicious cycle that accelerates β-cell dysfunction and the onset of T2D.

The topic is relevant is relevant to the field of beta cell physiology and up to date. I have only few suggestions that can improve the text

Comment 1. Figure 2. Please reconsider the cartoon (not clear the Trx function in H2O2 handling) and describe panel C in the figure legend. According to the authors’ idea and the cartoon (figure 2), trx function mainly in the cytosol and ERdj5 in the ER. Is this correct? Please better explain this point in the text and figure legend.

Author response: We apologize for this lack of clarity. We have amended Fig. 2 to better align with our descriptions in the text. We have also added additional language to the text regarding the electron shuttle from cytosolic thioredoxin to ER lumenal PDIs, such as ERdj5, to more clearly describe the current knowledge regarding this pathway. The following text was added to Page 5, line 226:

“Thus, how cytosolic thioredoxin contributes to the resolution of non-native disulfide bonds in the ER lumen is not clear. In prokaryotes, thioredoxin is restricted to the cytoplasm yet critical for protein disulfide bond reduction within the periplasmic space. Electron transfer occurs through reduction of the membrane protein DsbD, which shuttles electrons to the periplasmic protein disulfide isomerase, DsbC for disulfide bond exchange64 . In mammalian cells, recent studies demonstrate a similar electron shuttle occurs between cytosolic thioredoxin and an ER membrane protein to support disulfide bond reduction in the ER lumen65; however, the identity of the membrane protein and distal steps for transfer to PDIs, such as ERdj5, have yet to be defined.”

Comment 2. The authors suggest that alterations in the redox state in the ER directly affect mitochondrial function via ER-mitochondrial contact sites. However, recent data indicate that MERCs are altered (decreased) in T2D (DOI: 10.1016/bs.ircmb.2021.06.001). How can this evidence be in line with the authors' hypothesis?

Author response: This is an excellent point. Early in the development of β-cell dysfunction there is evidence of increased MERC number and function. This is consistent with acute ER stress models. However, as pointed out by the reviewer, studies of β-cells from advanced T2D models have shown decreased MERCs, which we propose is likely a consequence of the IP3R-mediated mitochondrial Ca2+ overload. The following text was added page 8, line 331:

“In the development of T2D, the extent of MERC tethering and/or the number of MERCs varies between stages of ER stress and β-cell dysfunction90. For example, acute glucose stimulation of human islets increased ER-mitochondrial tethering and Ca2+ exchange91, similar to acute activation of ER stress with tunicamycin86. Tentatively, this protective mechanism operates to improve organelle bioenergetics and functions. In contrast, islet studies using chronic culture with elevated glucose demonstrated depletion of ER Ca2+ stores and increased mitochondrial Ca2+ that coincided with defects in mitochondrial function, ATP generation, and overall diminished glucose-stimulated insulin secretion (GSIS)91. Thus, excess ER-mitochondria Ca2+ exchange can contribute to organelle dyshomeostasis and damage. In support of this notion, mitochondrial calcium overload through IP3R was reported in human T2D patients, mouse models of diabetes, and cultured β-cells92. In these advanced models of β-cell dysfunction, MERC numbers are diminished, which further highlights the complex and dynamic regulation of MERC assembly and function in the β-cell response to metabolic perturbations.”